# A Convenient Plant-Based Detection System to Monitor Androgenic Compound in the Environment

**DOI:** 10.3390/plants8080266

**Published:** 2019-08-05

**Authors:** Dong-Gwan Kim, Ramin Bahmani, Jae-Heung Ko, Seongbin Hwang

**Affiliations:** 1Department of Bioindustry and Bioresource Engineering, Department of Molecular Biology and Plant Engineering Research Institute, Sejong University, Seoul 05006, Korea; 2Department of Plant & Environmental New Resources, Kyung Hee University, Kyeonggi-do 17104, Korea

**Keywords:** endocrine disruptor, dihydrotestosterone, anthocyanin, *PtrMYB119*, *Arabidopsis thaliana*, androgen receptor

## Abstract

Environmental androgen analogues act as endocrine disruptors, which inhibit the normal function of androgen in animals. In the present work, through the expression of a chimeric gene specified for the production of the anthocyanin in response to androgen DHT (dihydrotestosterone), we generated an indicator *Arabidopsis* that displays a red color in leaves in the presence of androgen compounds. This construct consists of a ligand-binding domain of the human androgen receptor gene and the poplar transcription factor gene *PtrMYB119*, which is involved in anthocyanin biosynthesis in poplar and *Arabidopsis*. The transgenic *Arabidopsis* XVA-*PtrMYB119* displayed a red color in leaves in response to 10 ppm DHT, whereas it did not react in the presence of other androgenic compounds. The transcript level of *PtrMYB119* peaked at day 13 of DHT exposure on agar media and then declined to its normal level at day 15. Expressions of anthocyanin biosynthesis genes including chalcone flavanone isomerase, chalcone synthase, flavanone 3-hydroxylase, dihydroflavonol 4-reductase, UFGT (UGT78D2), and anthocyanidin synthase were similar to that of *PtrMYB119*. It is assumed that this transgenic plant can be used by nonscientists for the detection of androgen DHT in the environment and samples such as food solution without any experimental procedures.

## 1. Introduction

Endocrine-disrupting compounds (EDCs) are known as chemicals that disrupt the endocrine (or innate hormone) systems in animals at certain doses by interfering with endocrine hormonal functionality. These disruptions can induce cancerous tumors, birth defects, and various developmental disorders [1]. In particular, EDCs may be associated with the development of neurological disorders [2], severe attention deficit disorder [3], cognitive and brain development problems [4], breast, prostate, thyroid, and other cancers [5], deformations of the body, sexual development disorders such as feminizing males or masculinizing females, etc. [6].

EDCs are also found in many household and industrial products—even cosmetics and personal care products [7]. To date, 8300 million metric tons (Mt) of new plastics have been produced [8]. Global factories now produce approximately 400 million metric tons of plastic per year, which is more than a billion kilograms per day [9]. It contaminates the environment, including the atmosphere, wastewater effluents, agricultural soils, and sewage sludge [10,11,12].

It is very difficult to identify and quantify chemical pollutants or EDCs present in the environment. To date, EDCs usually have been detected using gas chromatography and mass spectrometry [13,14,15]. However, these devices are not easily available to the general public, and there are many limitations that prevent non-scientists from using such equipment. Because of this, the development of biomarkers to detect EDCs has been attempted. For example, a vitellogenin was developed as a EDC biomarker that interacts with the estrogen receptor (ER) in teleost species such as cyprinids or salmonids [16,17,18], and the glue protein spiggin has been used as an (anti)-androgenic compound marker in the three-spined stickleback *Gasterosteus aculeatus* [19,20,21].

The first anti-androgenic activities in the environment were quantified in the water and the sediment of a polluted river in Italy, which contained 1.34 μM (0.54 ppm) and 17.1 μM (6.84 ppm) FLU (flutamide) equivalents, respectively [22]. In addition, the European river had a total androgenic potency of 3820 pM dihydrotestosterone equivalents/g dry weight (1.11 ppm) in a total sediment extract [23].

During the last years, several studies have been carried out to generate transgenic *Arabidopsis* plants that detect estrogenic chemicals in the environment. All these experiments take advantage of a chimeric construct, including the hER-LBD (human estrogen receptor ligand-binding domain), LexA-DBD (LexA DNA-binding domain from *Escherichia coli*), and a nuclear receptor coactivator consisting of the hTIF2-NID (nuclear receptor interaction domain of human transcriptional intermediary factor 2) and VP16-AD (transactivation domain of VP16 from the herpes simplex virus) [24]. Using this approach, Tojo et al. (2006) [24] generated transgenic *Arabidopsis* lines in which the transcription of a marker gene (GUS; β-glucuronidase) is induced by various estrogenic compounds such as estrogen (17β-estradiol, at concentrations as low as 50 pM), genistein (at >100 nM), and nonylphenol (at 1 µM). Moreover, GFP-expressing transgenic *Arabidopsis* plants have been developed that exhibit a green fluorescence signal at the root following 17β-estradiol, BPA (bisphenol A), and NP (nonyl phenol) treatments [25]. Very recently, we generated transgenic *Arabidopsis* lines that synthesize anthocyanin in leaves in response to 10 ppm BPA [26].

Anthocyanins are color pigments that play a vital role in plants. They are involved in UV protection, plant–microbe interaction, male fertility, and response to nutrient availability [27,28]. Anthocyanin biosynthesis has been studied extensively, and detailed synthetic pathways are well known. Anthocyanin biosynthesis is controlled by the action of R2R3-MYB, basic helix-loop-helix (bHLH), and WD40 [MYB–bHLH–WD40 TF complex (MBW)] transcription factors that regulate the expression of the genes involved in this pathway [29,30,31]. Recently, *PtrMYB119* was isolated from poplar (*Populus trichocarpa*), which is an R2R3-MYB transcription factor and homologous to *Arabidopsis* PAP1 (production of anthocyanin pigment 1). Overexpression of *PtrMYB119* has been shown to increase anthocyanin production by stimulating the transcription of the anthocyanin biosynthesis genes in *Arabidopsis* and poplar [32]. 

Although several groups employed the XVE (X: DNA-binding domain of LexA; V: transcriptional activation domain of VP16; E: ligand-binding domain of the human estrogen receptor) inducible system in *Arabidopsis* to detect the estrogenic compound [24,25], they did not provide a convenient method, since established experimental biology knowledge as well as tools are essential in the provided procedures. Even if the color is expressed in roots, it cannot be observed with the naked eye if the plants were raised in soil. In the present work, a transgenic *Arabidopsis* plant was developed that produces anthocyanin in leaf (resulting in a red color) when exposed to DHT (dihydrotestosterone), an androgen compound, as a result of *PtrMYB119*-induced anthocyanin biosynthesis.

## 2. Results and Discussion

### 2.1. Preliminary Evaluation of the Transgenic Arabidopsis for DHT Detection

We developed a plant system to detect androgenic compounds. After transformation of the wild-type *Arabidopsis* (Col-0) with a chimeric construct carrying a human androgen receptor (see Materials and Methods for details), 26 transgenic lines were obtained based on Basta-resistance selection. All the transgenic lines were examined for their color inducibility in response to 10 ppm DHT. Finally, two homozygous lines (XVA-6 and XVA-8, T_3_) were selected for further experiments.

### 2.2. Anthocyanin Production in Transgenic Arabidopsis Is Influenced by Expression Levels of PtrMYB119 and Anthocyanin Biosynthesis Genes after DHT Treatment

To examine the sensitivity of the XVA-*PtrMYB119* transgenic plants to detect androgenic compounds, two homozygous lines (XVA-6 and XVA-8) were germinated and grown on 1/2 MS (Murashige-Skoog) agar medium containing different concentrations of DHT (0–10 ppm), and their leaf color change was monitored. Leaf color change was evident in the transgenic plants at 1 ppm DHT, whereas the leaves of wild-type plants (Col-0) did not show any red color change (Figure 1).

The anthocyanin level enhanced in response to DHT from 1 ppm, reaching a 5-fold enhancement at 10 ppm DHT (Figure 2A). In support of this, the transcript level of PtrMYB119 was increased by DHT from 1 ppm, reaching a 3.8-fold level at 10 ppm DHT (Figure 2B). To explain the induction of anthocyanin by PtrMYB119 expression, the expressions of AtCHS (chalcone synthase) and AtANS (anthocyanidin synthase), which are rate-limiting first and last step genes in the anthocyanin biosynthesis pathway, were measured. As shown in Figure 2C,D, transcript levels of these genes were increased from 1 ppm DHT, as in the anthocyanin level and the PtrMYB119 expression. In contrast, the induction in the transcript levels of these genes was not observed in wild-type (Col-0) plants following DHT treatment. These patterns of XVA-PtrMYB119 plants were different from those of transgenic *Arabidopsis* XVE-PtrMYB119, in which PtrMYB119 expression began increasing even at 1 ppt BPA, whereas the anthocyanin level and the expressions of anthocyanin biosynthesis genes increased at 10 ppm BPA [26]. This difference may be attributed to the different receptor binding affinity of BPA and DHT. Furthermore, XVA-PtrMYB119 plants did not exhibit increases in red color, anthocyanin contents, or PtrMYB119 expression in response to the androgenic compounds vinclozolin and methyltestosterone due to an as yet unknown reason (data not shown).

### 2.3. Time-Dependent Anthocyanin Synthesis in Response to DHT in TRANSGENIC ARABIDOPSIS

After growing the transgenic plants on the media containing 10 ppm DHT, the leaf color started changing on day 12 (D12), maximized on day 13 (D13), and disappeared by day 15 (D15) (Figure 3). The anthocyanin level and the *PtrMYB119* expression were in accordance with the color change pattern, showing the maximum level at D13 and the minimum level at D14 or D15 (Figure 4A,B). However, in the control plants (Col-0), the red color in leaf, anthocyanin content, and *PtrMYB119* expression were not enhanced by DHT (Figure 3 and Figure 4).

The transcript levels of anthocyanin biosynthetic genes *AtCHS*, *AtCHI* (encoding chalcone flavanone isomerase, AT3G55120), *AtF3H* (encoding flavanone 3-hydroxylase, AT3G51240), *AtDFR* (encoding dihydroflavonol 4-reductase, AT5G42800), *AtANS*, and *AtUFGT* (encoding UGT78D2, AT5G17050) were matched with patterns of red color, anthocyanin content, and *PtrMYB119* expression, peaking at day 13 (D13) and reducing to the lowest levels at day 15 (D15) (Figure 4C–H). Again, these patterns of XVA-*PtrMYB119* plants were different from those of XVE-*PtrMYB119* plants, since expressions of *PtrMYB119* and anthocyanin biosynthesis genes peaked at D11, which was one day before the maximum level of anthocyanin (red color) at D12 in XVE-*PtrMYB119* plants [26].

Based on these results, it is assumed that the anthocyanin production in transgenic plant leaves is highly dependent on the expression level of *PtrMYB119*. Regarding the transcript level of *PtrMYB119*, which maximized on D13 and reduced to the minimal level at D14 or D15, it is suggested that its expression was switched off after D13 by an unknown mechanism. Moreover, the possible degradation of DHT or its conversion to other compounds as in BPA [24] cannot be excluded.

Over the past decades, several systems have been developed to detect EDCs using human cells, yeast, and reporter genes [24,25,33,34]. However, these systems are inconvenient for the general public to use. For example, sterilization conditions should be maintained, or certain specific reagents or equipment should be used. On the other hand, an EDC detection system using plant color change is more convenient because it is easy to store and use plants, and it does not require the use of specific reagents or equipment. In addition, previously reported transgenic *Arabidopsis* expressed reporters only in roots [24,25], which are very difficult to observe if plants are grown in soil. Until now, a plant system to detect androgenic compounds has never been developed. Taken together, the generated transgenic plants in this study confer an easy-to-use system to detect DHT (an androgen) in the environment. This is achieved through the red color change in the leaves, which can be observed by the naked eye without any additional staining or microscopic observation.

Since our transgenic plants cannot detect DHT at levels lower than 10 ppm, the sensitivity of XVA-*PtrMYB119* plants to DHT should be improved. First, the transcriptional complex for the anthocyanin synthesis MYB–bHLH–WD40 TF complex [29,30,31] could be activated by overexpressing the *Arabidopsis* genes of these components. Second, a stronger promoter than the 35S CaMV promoter could be used to increase the synthesis of the androgen receptor and thus of *PtrMYB119*.

## 3. Materials and Methods

### 3.1. Plant Material, Growth Condition, and Chemical Treatments

All the experiments were performed using wild-type *Arabidopsis* plants (Columbia-0 ecotype, stock number CS70000). Seed sterilization and growth conditions were followed as previously described [35]. For chemical treatments, DHT (5α-Androstan-17β-ol-3-one C-IIIN, Sigma-Aldrich, Cat# A8380), vinclozolin (Sigma-Aldrich, Cat# 45705), and methyltestosterone (Sigma-Aldrich, Cat# 46444) were dissolved in DMSO to make a 100 mM stock solution, which was added to 1/2 MS media after autoclaving to prepare the indicated concentrations (from 1 ppt to 10 ppm).

### 3.2. Plasmid Construction and Plant Transformation

Chimeric gene construction harboring an XVA (X: DNA-binding domain of LexA; V: transcriptional activation domain of VP16; A: ligand-binding domain of the human androgen receptor) segment was generated as previously described [26,36]. The XV region was amplified from the pMDC7 vector template, and the A region was amplified from the NM_00044.2 template. PCR fragments were fused by overlapping PCR using XV_*Spe*I_F (forward XV primer sequences with 5’ *Spe*I linker), XV_pA_R (reverse XV primer sequences with 5’ androgen receptor gene-specific linker), pXV_A_F (forward androgen primer sequences with 3’ XV gene-specific linker), and A_*Spe*I_R (reverse androgen primer sequences with 3’ *Spe*I linker). Finally, an XVA fragment with a forward and reverse *Spe*I restriction sequence was cloned into pB2GW7 using the Gateway system following the manufacturer’s guide (Invitrogen, Carlsbad, CA, USA). The resulting construct was validated by DNA sequencing. Consequently, the *Arabidopsis* wild-type plant (Col-0) was transformed with *Agrobacterium tumefaciens* (strain GV3101) harboring the designated vector [37]. Transformants were selected on the Basta medium (5 µg/mL). The primer sequences used for gene cloning are given in the Supporting Information (Appendix A).

### 3.3. RNA Isolation and Quantitative Real-Time PCR (qRT-PCR)

RNA isolation, cDNA synthesis, and qRT-PCR were performed as described previously [38]. Total RNA was isolated from a transgenic *Arabidopsis* seedling (T3 homozygous) using an IQeasy plus Plant RNA extraction kit (Intron, Korea). After quantification of the RNA using NanoDrop (BioSpec-nano, Shimadzu, Japon), first-strand cDNA was synthesized from 2 μg of RNA in RNase-free water using a PrimeScript™ RT reagent kit (Takara, Japan) according to the manufacturer’s instructions. The qRT-PCR was carried out using a CFX Connect™ Real-Time PCR Detection System (Bio-Rad Laboratories Inc., Hercules, CA, USA). The reaction mixture consisted of 10 μL of SYBER Supermix (SsoAdvanced™ Universal SYBR^®^ Green Supermix, Bio-Rad Laboratories Inc., Hercules, CA, USA), 1 μL of cDNA, 7 μL of nuclease-free water, and 1 μL of each primer to a final volume of 20 μL. The qRT-PCR reaction thermal conditions were as follows: 95 °C for 30 s, 40 cycles at 95 °C for 15 s, and 57 °C for 20 s. The transcript level of the genes was normalized to the transcript level of AtActin2 as a reference gene. Experiments were carried out using three independent biological samples, and each qRT-PCR reaction included three technical replications. The sequences of the primers are given in the Supporting Information (Appendix A).

### 3.4. Anthocyanin Measurement

In order to quantify the anthocyanin content, the method of Nakata et al. [39] was adopted. Briefly, plant samples were ground in LN_2_ (liquid nitrogen) and homogenized in assay buffer (containing 45% (*v/v*) cold methanol and 5% (*v/v*) acetic acid solution). The supernatant obtained by centrifugation (10,000× *g*) was utilized for anthocyanin measurement. The optical density of the samples was determined at 530 and 657 nm using a spectrophotometer (SP-2000UV). The anthocyanin concentration was calculated based on the following equation: [OD530-(0.25 × OD657)] × extraction volume (mL) × 1/weight of a tissue sample (g).

### 3.5. Light Microscopy Analysis

A Leica microscope (MZ10F, Buffalo Grove, IL, USA) equipped with a digital camera (DFC450) was employed for photography of the leaves under 10× magnification. The experiments were performed in triplicate, and 10 plants were photographed per line; a representative image is shown for each line.

### 3.6. Statistical Analysis

All the experiments were performed in triplicate, and the mean is represented in the graphs. Data were analyzed with SAS software (version 9.1), and the difference between the means was determined using Tukey’s HSD (honestly significant difference) test at *p* ≤ 0.05.

## 4. Conclusions

We developed transgenic *Arabidopsis* lines that produce a red color in leaf in response to the androgen hormone DHT. This response is caused by the expression of *PtrMYB119*, which is induced by the binding of androgenic components to the androgen receptor, leading to anthocyanin biosynthesis in the abaxial surface of the leaf. These plant lines provide a fast, convenient, and cost-effective system to detect the presence of DHT in testing materials. Because no additional experimental procedures are required, it is assumed that even non-scientists will be able to detect the androgen DHT in soils and various test samples such as drinking water through the use of the proposed method.

## Figures and Tables

**Figure 1 plants-08-00266-f001:**
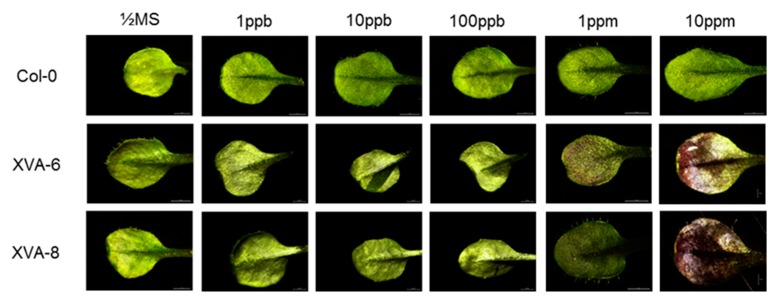
Anthocyanin production in leaves of transgenic XVA-*PtrMYB119 Arabidopsis* by DHT (dihydrotestosterone) treatment. The seeds were germinated on 1/2 MS agar medium containing different concentrations of DHT (0–10 ppm) and grown for 14 days.

**Figure 2 plants-08-00266-f002:**
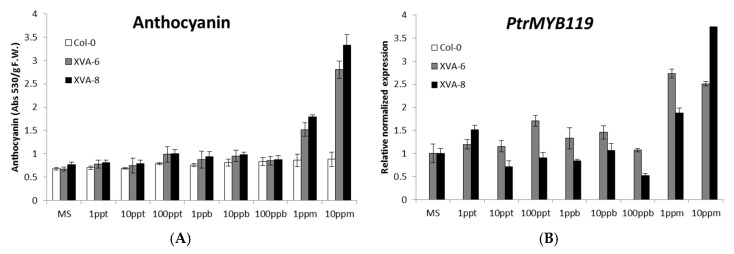
Concentration-dependent experiments exhibiting the anthocyanin content and the expression levels of genes involved in anthocyanin biosynthesis after DHT treatment. (**A**) Anthocyanin content in wild-type (Col-0) and transgenic *Arabidopsis* plants. The seeds were germinated on 1/2 MS agar medium containing different concentrations of DHT (0–10 ppm) and grown for 14 days. (**B**–**D**) qRT-PCR analyses of *PtrMYB119*, *AtCHS*, and *AtANS* expressions in wild-type (Col-0) and transgenic *Arabidopsis* plants. Experiments were performed using seedlings on day 13, at which point the highest anthocyanin synthesis was observed. All experiments were performed in triplicate, and the average is represented in the graphs.

**Figure 3 plants-08-00266-f003:**
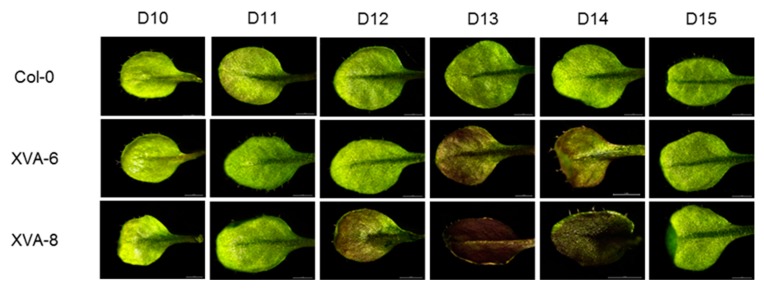
Leaf color changes in transgenic XVA-*PtrMYB119 Arabidopsis* by 10 ppm DHT treatment. The seeds were germinated on 1/2 MS agar medium in the presence of 10 ppm DHT and grown for 15 days. Leaf color change was examined daily after seed germination (D: day).

**Figure 4 plants-08-00266-f004:**
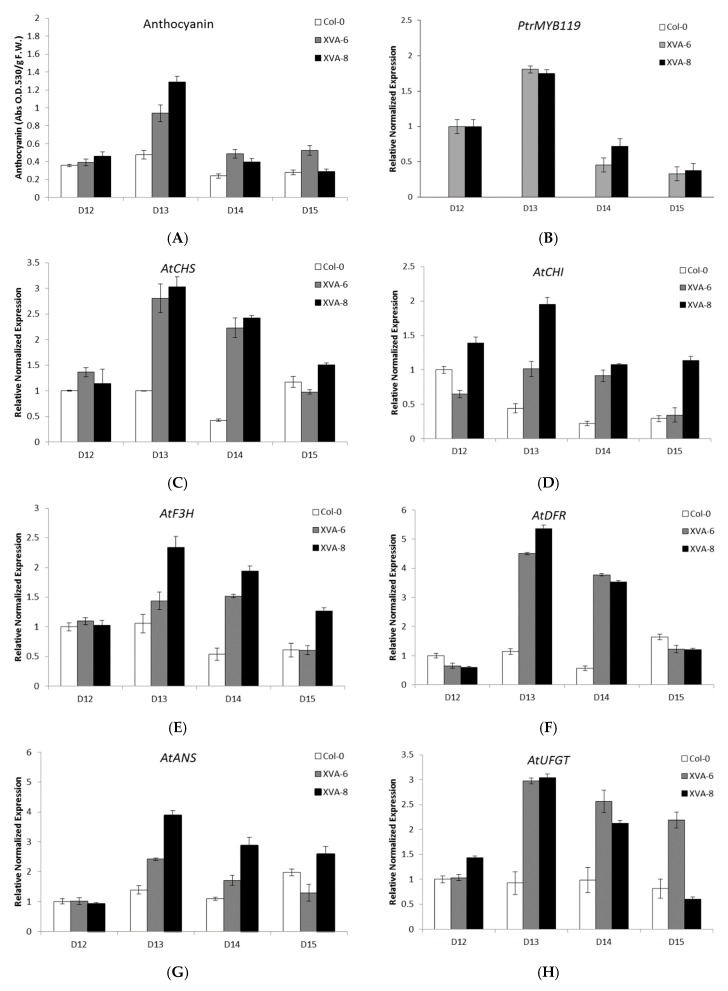
Time course experiments showing the anthocyanin content and the expression levels of anthocyanin biosynthesis genes in wild-type (Col-0) and transgenic *Arabidopsis* plants. (**A**) Anthocyanin contents in wild-type (Col-0) and transgenic *Arabidopsis*. The seeds were germinated on 1/2 MS agar medium containing 10 ppm DHT and grown for 15 days. (**B**–**H**) The transcript level of anthocyanin biosynthesis genes was analyzed by qRT-PCR. *AtActin2* was used as an internal control. Experiments were performed using seedlings at 12–15 days after seed germination (D: day). All experiments were performed in triplicate, and the average is represented in the graphs.

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
