# Peer review of "A Convenient Plant-Based Detection System to Monitor Androgenic Compound in the Environment"

_plants, 2019, doi:10.3390/plants8080266_

Round 1
Reviewer 1 Report
This article describes a transgenic plant based system for facile DHT (dihydrotestosterone) detection by the plant producing of anthocyanin and the subsequent appearance of a red colour in the Arabidopsis leaves.
The article is well written although somewhat short coming close to being a communication. The authors must include conclusions and future perspectives.
A glossary of terms and abbreviations would be useful especially to the reader who is not directly an expert in this field.
Author Response
Dear reviewer,
I appreciate your valuable comments.
1. I revised and received English editing, so you can find them in colors in a manuscript.
2. I changed abbreviations to full names as many as possible.
3. I added future directions/improvements and conclusion as follows.
Since our transgenic plants cannot detect DHT at levels lower than 10 ppm, the sensitivity of XVA-PtrMYB119 plants to DHT should be improved. First, the transcriptional complex for the anthocyanin synthesis MYB–bHLH–WD40 TF complex [29-31] (29–-31) could be activated by overexpressing the Arabidopsis genes of these components. Second, a stronger promoter than the 35S CaMV promoter could be used to increase the synthesis of the androgen receptor and thus of PtrMYB119 .
3. Conclusions
We developed transgenic Arabidopsis lines that produce a red color in leaf in response to the androgen hormone DHT. This response is caused by the expression of PtrMYB119, which is induced by the binding of androgenic components to the androgen receptor, leading to anthocyanin biosynthesis in the abaxial surface of the leaf. These plant lines provide a fast, convenient, and cost-effective system to detect the presence of DHT in testing materials. Because no additional experimental procedures are required, it is assumed that even non-scientists will be able to detect the androgen DHT in soils and various test samples such as drinking water through the use of the proposed method.
Reviewer 2 Report
Comments:
In this study, the outcome revealed that the anthocyanin production in transgenic plant leaves is highly dependent on the expression level of PtrMYB119. Regarding the transcript level of PtrMYB119 which maximized on D13 and reduced to the minimal level at D14 or D15. Also, the authors suggested that its expression was switched off after D13 by an unknown mechanism. The article is very well written, and it has a nice flow of information. It is very easy to read and understand.
Page 2 Line 39: Provide the abbreviation ……DCs……
Page 2 Line 53: Provide the abbreviation ……FLU……
Page 2 Line 65: Provide the abbreviation ……GFP……
Page 2 Line 66: Provide the abbreviation ……NP……
Page 8, Line 255: Authors should provide the details of RNA isolation and what type of kit used for the study in details. Also, authors can mention the details of the qRT-PCR reaction mixture, components and cycle details.
Authors could improve the quality of the discussion in the manuscript; the overall discussion of this manuscript is a very week; it could be improved.
Author Response
Dear reviewer,
I appreciate your valuable comments.
I revised and received English editing, so you can find them in re-submitted manuscript. I changed abbreviated words to full names. I revised qRT-PCR section as follows.RNA isolation, cDNA synthesis, and qRT-PCR were performed as described previously [38]. Total RNA was isolated from a transgenic Arabidopsis seedling (T3 homozygous) using an IQeasy plus Plant RNA extraction kit (Intron, Korea). After quantification of the RNA using NanoDrop (BioSpec-nano, Shimadzu, Japon), first-strand cDNA was synthesized from 2 μg of RNA in RNase-free water using a PrimeScript™ RT reagent kit (Takara, Japan) according to the manufacturer’s instructions. The quantitative reverse transcriptase (qRT-PCR) was carried out using a CFX Connect ™ Real-Time PCR Detection System (Bio-Rad Laboratories Inc., USA). The reaction mixture consisted of 10 μl of SYBER Supermix (SsoAdvanced™ Universal SYBR® Green Supermix), 1 μl of cDNA, 7 μl of nuclease-free water, and 1 μl of each primer to a final volume of 20 μl. The qRT-PCR reaction thermal conditions were as follows: 95 °C for 30 s, 40 cycles at 95 °C for 15 s, and 57 °C for 20 s. The transcript level of the genes was normalized to the transcript level of AtActin2 as a reference gene. Experiments were carried out using three independent biological samples and each qRT-PCR reaction includes three technical replications. The sequences of the primers are given in the Supporting Information (Table S1). I improved Discussion as follows. Since our transgenic plants cannot detect DHT at levels lower than 10 ppm, the sensitivity of XVA-PtrMYB119 plants to DHT should be improved. First, the transcriptional complex for the anthocyanin synthesis MYB–bHLH–WD40 TF complex could be activated by overexpressing the Arabidopsis genes of these components. Second, a stronger promoter than the 35S CaMV promoter could be used to increase the synthesis of the androgen receptor and thus of PtrMYB119 .
Reviewer 3 Report
Manuscript entitled "A convenient plant-based detection system to monitor androgenic compound in the environment" is within the scope of Plants journal. It is good experimental article with interesting subject and good experimental work. The manuscript is fairly well written and includes a great deal of information, which is reflected in the significant number of references listed. The methodology of experimental part is well established and it does not raise any objections. The results and discussion are represented in a logical way. Although in Discussion section there is no comparison with the literature and in my opinion this should be corrected. Authors included 4 figures which are clear and legible.
The manuscript should be careful proof read in terms of typos and language. English language should be substantially improved by a native English speaker.
Author Response
Dear reviewer,
I appreciate your valuable comments,
I revised and received English editing, so you can see them in colors in the re-submitted manuscript. I improved Discussion as follows.
Since our transgenic plants cannot detect DHT at levels lower than 10 ppm, the sensitivity of XVA-PtrMYB119 plants to DHT should be improved. First, the transcriptional complex for the anthocyanin synthesis could be activated by overexpressing the Arabidopsis genes of these components. Second, a stronger promoter than the 35S CaMV promoter could be used to increase the synthesis of the androgen receptor and thus of PtrMYB119 .
Conclusions
We developed transgenic Arabidopsis lines that produce a red color in leaf in response to the androgen hormone DHT. This response is caused by the expression of PtrMYB119, which is induced by the binding of androgenic components to the androgen receptor, leading to anthocyanin biosynthesis in the abaxial surface of the leaf. These plant lines provide a fast, convenient, and cost-effective system to detect the presence of DHT in testing materials. Because no additional experimental procedures are required, it is assumed that even non-scientists will be able to detect the androgen DHT in soils and various test samples such as drinking water through the use of the proposed method.